# Cloacal Dysgenesis Sequence

**DOI:** 10.3390/diagnostics13233529

**Published:** 2023-11-24

**Authors:** Nicolae Gică, Livia Apostol, Iulia Huluță, Corina Gică, Nicoleta Gana, Ana-Maria Vayna

**Affiliations:** 1Obstetrics and Gynecology Department, Faculty of Medicine, “Carol Davila” University of Medicine and Pharmacy, 050474 Bucharest, Romania; gica.nicolae@umfcd.ro (N.G.); iuliahuluta16@gmail.com (I.H.); gana_nicoleta@yahoo.com (N.G.); 2Department of Obstetrics and Gynecology, Filantropia Clinical Hospital Bucharest, 011132 Bucharest, Romania; mat.corina@gmail.com (C.G.); anamariaiane@yahoo.com (A.-M.V.)

**Keywords:** cloacal dysgenesis sequence, fetal obstructive uropathy, keyhole sign

## Abstract

This article presents a rare case of cloacal dysgenesis sequence (CDS) detected at 23 weeks of gestation in a 36-year-old woman’s first ongoing pregnancy. The fetal ultrasound demonstrated anhydramnios, megacystis, the “keyhole sign” and empty bilateral renal fossae, findings consistent with the fetal obstructive uropathy (FOU). A subsequent postmortem carried out confirmed a diagnosis of a cloacal dysgenesis sequence, characterized by the absence of anal, genital and urinary openings with intact perineum covered by smooth skin and a phallus-like structure.

Cloacal dysgenesis sequence (CDS) is a rare cause of fetal obstructive uropathy (FOU) and seen in only 1:50,000 to 250,000 pregnancies [1]. This congenital anomaly is typically incompatible with long-term survival and leads to the development of early oligohydramnios, which subsequently results in fatal pulmonary hypoplasia. The bladder outlet obstruction and pulmonary hypoplasia due to the lack of amniotic fluid and/or kidneys abnormalities should be considered as prognostic factors for neonatal death [2]. Although CDS occurs mainly in females, there have been reported cases of cloacal dysgenesis sequence (CDS) detected in males as well [3]. The prenatal differential diagnosis of cloacal dysgenesis sequence from other urinary obstructive diseases is difficult, but very important for evaluating fetal prognosis and guiding both prenatal and neonatal care [4].

We report a rare case of a 36-year-old woman referred to our fetal medicine department due to anhydramnios detected at 23 weeks of pregnancy. This was her first ongoing pregnancy following spontaneous conception. The patient is known with three previous missed miscarriages. The first trimester combined screening tests for common chromosomal conditions, based on a combination of maternal age, ultrasound findings (fetal nuchal translucency, fetal heart rate, fetal nose, the flow of blood across the tricuspid valve of the fetal heart and the ductus venosus) and serum-free ß-hCG and PAPP-A, which demonstrated a low chance result for Trisomy 21 and Trisomies 13/18. The first trimester scan, performed in a private practice, showed fetal megacystis, hypoplastic nasal bone, micrognathia and reversed flow in the ductus venosus with a nuchal translucency measurement of 2 mm. The cell-free DNA test was performed and the result demonstrated a low risk for common chromosomal abnormalities with a fetal fraction of 7%.

The ultrasound scan in our fetal medicine department, performed at 23 weeks, demonstrated the following abnormalities in the fetus (Figure 1):Fetal megacystis: increased bladder wall thickness, bladder dilatation and the presence of the ‘keyhole sign’.Empty bilateral renal fossae with adrenal glands parallel to the spine.Single umbilical artery.Anhydramnios.Positional fetal scoliosis and positional bilateral talipes.Pulmonary hypoplasia.

The combination of abnormalities observed in this case was considered to be consistent with the fetal obstructive uropathy. The prognosis for the fetus is concerning due to the lack of amniotic fluid, which would significantly affect lung development and the perinatal mortality is secondary to respiratory failure. A crucial stage of lung development is between 16 and 20 weeks, which is the canalicular phase, and a lack of adequate amniotic fluid is affecting this. These findings are associated with a poor prognosis and a high risk of perinatal demise. The options of management were discussed, which include further investigations and tests with expectant management or a termination of pregnancy. Unfortunately, the patient declined the amniocentesis for karyotyping or other investigations. The option of the termination of pregnancy was offered, given the presence of the anhydramnios—a poor prognostic marker in a urethral obstruction abnormality.

A subsequent postmortem carried out confirmed the characteristic primary malformation sequence of cloacal dysgenesis sequence (CDS), which, in this case, included absent anal, genital and urinary orifices, a smooth median raphe, urethral atresia and a phallus-like structure. Also, the kidneys were small and dysplastic without corticomedullary differentiation, the ureters were not identified and the bladder was very distended and filled with meconium with rectovesical fistula and urethral atresia.

In conclusion, this case report provides us with a perspective on the intricacies involved in diagnosing the cloacal dysgenesis sequence (CDS), which presents challenges in both prenatal diagnosis as well as during pathological examination. It is important to distinguish CDS from malformations such as fetal obstructive uropathies, which could potentially benefit from intrauterine surgical intervention and are associated with a better overall prognosis.

## Figures and Tables

**Figure 1 diagnostics-13-03529-f001:**
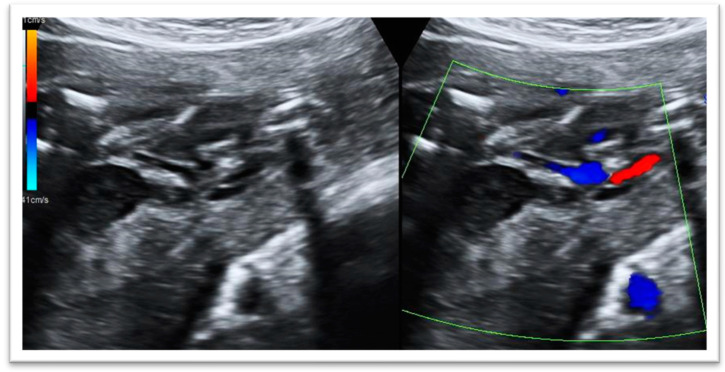
Assessment of the fetal bladder: *the keyhole sign* is an ultrasonographic marker which refers to the appearance of the dilatation of the proximal urethra and a thick-walled distended bladder and single umbilical artery.

## Data Availability

Not applicable.

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
