# Peer review of "Cloacal Dysgenesis Sequence"

_diagnostics, 2023, doi:10.3390/diagnostics13233529_

Round 1
Reviewer 1 Report
Comments and Suggestions for Authors
1. Data should be added on when (at which gestational age) the tests and scans were made
2. Clarify: was the 1st trimester screening test the PAPP-A test?
3. There is only one figure with 2 scans - I suggest to illustrate this case with more US scans demonstrating other abnormalities found in this fetus
Comments on the Quality of English LanguageGeneral quality of English language in this manuscript is good. Minor corrections are required: for example: "non-consanguineous marriage" (should be non-consanguineous parents)
Author Response
Thank you for your feedback on our manuscript.
Please find below the responses to comments and suggestions regarding our publication.
- Data should be added on when (at which gestational age) the tests and scans were made
We appreciate your suggestion to include specific information about the gestational age at which the tests and scans were conducted. We acknowledge the importance of providing precise details regarding the timeline of these assessments. In our revised manuscript, we will incorporate the gestational age at which each test and scan was performed, providing a comprehensive timeline for clarity and accuracy. The ultrasound scan in our fetal medicine department was performed at 23 weeks.
- Clarify: was the 1st trimester screening test the PAPP-A test?
We apologize for any confusion caused by the lack of clarity in identifying the specific test. To address this, we will provide a clear clarification in the revised manuscript that the 1st trimester screening test for trisomies 21, 18 and 13 is based on a combination of maternal age, ultrasound findings ( fetal nuchal translucency, fetal heart rate, fetal nose, the flow of blood across the tricuspid valve of the fetal heart and the ductus venosus) and serum free ß-hCG and PAPP-A .
- There is only one figure with 2 scans - I suggest to illustrate this case with more US scans demonstrating other abnormalities found in this fetus.
Regrettably, due to the presence of anhydramnios during the fetal evaluation, obtaining high-quality static ultrasound images depicting the identified abnormalities beyond those represented in the figure was challenging. The rest of the anomalies were discovered in real-time assessment and were not captured with sufficient clarity to provide relevant static pictures for demonstration purposes.We remain committed to providing a thorough and accurate representation of our findings within the constraints of the available data. If there are alternative ways or supplementary information that could effectively convey the real-time assessment process or further details about the identified abnormalities, we would greatly appreciate your guidance in this matter.
The manuscript was reviewed accordingly.
Thank you once again for your attention and consideration. I look forward to further collaboration and guidance in refining this submission.
Best regards,
Dr. Nicolae Gica
Obstetrics and Gynaecology Department
Filantropia Clinical Hospital
Bucharest, Romania
Reviewer 2 Report
Comments and Suggestions for Authors
The case is interesting and of clinical value.
The discussion on potential diagnostic and therapeutic tools should be expanded - the usual way of presenting clinical cases contains review of the literature and it is advisable to add it in this case.
Are you sure that the informed consent was taken from the patient?
Comments on the Quality of English LanguageGrammatical errors should be corrected.
Some terms require verification, e.g. missed miscarriage?
Author Response
Thank you for your attention.
I appreciate the opportunity to clarify the intent behind the changes made to the paper.
The decision to categorise the paper under "Interesting Images" was deliberate and stemmed from the belief that the visual content within the research holds significant value and interest beyond the confines of a traditional case report. The revised title was chosen to accurately reflect the unique perspective and contribution of the images presented.
Regarding your concern about the absence of a literature review section, I want to highlight that the focus of this submission is to emphasize the visual aspects and their relevance. While a comprehensive literature review often enriches scientific discourse, the intention here is to spotlight the imagery and its inherent value without overshadowing it with extensive textual content.
We meticulously followed institutional protocols and ethical guidelines to obtain informed consent from the patient involved in this case. To safeguard patient confidentiality and privacy, specific details regarding the consent process were documented in accordance with institutional regulations.
I am open to discussing further suggestions or potential adjustments that could enhance the manuscript within the context of its intended category.
Thank you once again for your attention and consideration.
I look forward to further collaboration and guidance in refining this submission.
Best regards,
Dr. Nicolae Gica
Obstetrics and Gynaecology Department
Filantropia Clinical Hospital
Bucharest, Romania